# 5′ Region Large Genomic Rearrangements in the *BRCA1* Gene in French Families: Identification of a Tandem Triplication and Nine Distinct Deletions with Five Recurrent Breakpoints

**DOI:** 10.3390/cancers13133171

**Published:** 2021-06-25

**Authors:** Sandrine M. Caputo, Dominique Telly, Adrien Briaux, Julie Sesen, Maurizio Ceppi, Françoise Bonnet, Violaine Bourdon, Florence Coulet, Laurent Castera, Capucine Delnatte, Agnès Hardouin, Sylvie Mazoyer, Inès Schultz, Nicolas Sevenet, Nancy Uhrhammer, Céline Bonnet, Anne-Françoise Tilkin-Mariamé, Claude Houdayer, Virginie Moncoutier, Catherine Andrieu, Ivan Bièche, Marc-Henri Stern, Dominique Stoppa-Lyonnet, Rosette Lidereau, Christine Toulas, Etienne Rouleau

**Affiliations:** 1Department of Genetics, Institut Curie, F-75248 Paris, France; sandrine.caputo@curie.fr (S.M.C.); Adrien.Briaux@curie.fr (A.B.); virginie.moncoutier@curie.fr (V.M.); catherine.andrieu78@sfr.fr (C.A.); ivan.bieche@curie.fr (I.B.); marc-henri.stern@curie.fr (M.-H.S.); dominique.stoppa-lyonnet@curie.fr (D.S.-L.); rosette.lidereau@curie.fr (R.L.); 2Institut Curie, PSL Research University, F-75005 Paris, France; 3Laboratoire d’Oncogénétique, Institut Claudius Regaud, IUCT-O, F-31059 Toulouse, France; Telly.Dominique@claudiusregaud.fr; 4Department of Neurosurgery, Boston Children’s Hospital, Boston, MA 02115, USA; julie.sesen@childrens.harvard.edu; 5Roche Innovation Center Basel (RICB), Roche Pharma Research and Early Development, CH-4052 Basel, Switzerland; Maurizio.ceppi@roche.com; 6Laboratoire de Génétique Constitutionnelle et INSERM U916 VINCO, Institut Bergonié, CEDEX, F-33076 Bordeaux, France; F.Bonnet@bordeaux.unicancer.fr (F.B.); N.Sevenet@bordeaux.unicancer.fr (N.S.); 7Laboratoire d’Oncogénétique Moléculaire, Département de Biologie du Cancer, Institut Paoli-Calmettes, F-13273 Marseille, France; BOURDONV@ipc.unicancer.fr; 8Department of Genetics, Pitié-Salpêtriere Hospital, Assistance Publique-Hopitaux de Paris, Sorbonne University, F-75013 Paris, France; florence.coulet@aphp.fr; 9Laboratoire de Biologie et de Génétique du Cancer, CLCC François Baclesse, INSERM 1079 Centre Normand de Génomique et de Médecine Personnalisée, F-14076 Caen, France; l.castera@baclesse.unicancer.fr (L.C.); a.hardouin@baclesse.unicancer.fr (A.H.); 10Service de Génétique Médicale, Unité de Génétique Moléculaire, CHU Nantes, F-44093 Nantes, France; capucine.delnatte@chu-nantes.fr; 11Centre de Recherche en Neurosciences de Lyon, INSERM, U1028, CNRS, UMR5292, Université de Lyon, F-69008 Lyon, France; sylvie.mazoyer@inserm.fr; 12Centre Paul Strauss, Laboratoire de Biologie Tumorale—Oncogénétique, F-67000 Strasbourg, France; ines.schultz@chru-strasbourg.fr; 13Biologie Clinique et Oncologique, Biologie Moléculaire—Centre Jean Perrin, F-63000 Clermont-Ferrand, France; Nancy.UHRHAMMER@cjp.fr; 14Institut de Cancérologie, 6 Avenue de Bourgogne, F-54519 Vandœuvre-lès-Nancy, France; ce.bonnet@chru-nancy.fr; 15Cancer Research Center of Toulouse (CRCT), Institut National de la Santé et de la Recherche Médicale (INSERM) UMR 1037, F-31000 Toulouse, France; Toulas.Christine@claudiusregaud.fr; 16Inserm U1245, UNIROUEN, Normandie University, Normandy Centre for Genomic and Personalized Medicine, F-76183 Rouen, France; claude.houdayer@chu-rouen.fr; 17Normandy Centre for Genomic and 41 Personalized Medicine, Department of Genetics, University Hospital, F-76183 Rouen, France; 18Faculty of Pharmaceutical and Biological Sciences, University of Paris, F-75006 Paris, France; 19Institut Curie, INSERM U830, DNA Repair and Uveal Melanoma (D.R.U.M.), PSL Research University, F-75005 Paris, France; 20Faculty of Medicine, University of Paris, F-75005 Paris, France; 21Department of Biology, Gustave Roussy, Université Paris-Saclay, F-94805 Villejuif, France

**Keywords:** large genomic rearrangement, *BRCA1* gene, large duplication, triplication, variant of unknown significance

## Abstract

**Simple Summary:**

Large genomic rearrangements in *BRCA1* consisting of deletions/duplications of one or several exons are complex events, often occurring in the 5′ region. We characterized 10 events in 20 families: one large triplication classified as benign and nine large deletions classified as pathogenic. The breakpoint localization will certainly help to further understand the chromatin structure in regions sensitive to rearrangement.

**Abstract:**

Background: Large genomic rearrangements (LGR) in *BRCA1* consisting of deletions/duplications of one or several exons have been found throughout the gene with a large proportion occurring in the 5′ region from the promoter to exon 2. The aim of this study was to better characterize those LGR in French high-risk breast/ovarian cancer families. Methods: DNA from 20 families with one apparent duplication and nine deletions was analyzed with a dedicated comparative genomic hybridization (CGH) array, high-resolution BRCA1 Genomic Morse Codes analysis and Sanger sequencing. Results: The apparent duplication was in fact a tandem triplication of exons 1 and 2 and part of intron 2 of *BRCA1*, fully characterized here for the first time. We calculated a causality score with the multifactorial model from data obtained from six families, classifying this variant as benign. Among the nine deletions detected in this region, eight have never been identified. The breakpoints fell in six recurrent regions and could confirm some specific conformation of the chromatin. Conclusions: Taken together, our results firmly establish that the *BRCA1* 5′ region is a frequent site of different LGRs and highlight the importance of the segmental duplication and Alu sequences, particularly the very high homologous region, in the mechanism of a recombination event. This also confirmed that those events are not systematically deleterious.

## 1. Introduction

Five to ten percent of breast and ovarian cancers are hereditary mainly due to pathogenic variants in the susceptibility genes, *BRCA1* and *BRCA2*, transmitted in an autosomal dominant mode. Since the discovery of the *BRCA* genes, more than 2000 distinct variants have been identified by extensive mutational analysis and listed in the French database [1]. The *BRCA1* gene (MIM 113705), altered in terms of susceptibility to breast and ovarian cancers with a very high penetrance rate [2,3], is composed of 22 coding exons and two alternative non-coding exons (exons 1a and 1b). The *BRCA1* genomic sequence spans 81 kb from chr17:43044295–43125483 (A of the ATG is at 43125483) in a telomeric-centromeric orientation. The *BRCA1* gene characteristically has an extremely high density of intronic Alu repeats and a duplicated promoter region containing a *BRCA1* pseudogene that most likely accounts for the occurrence of “hot spots” that favor unequal homologous recombination events [4,5]. These unequal homologous recombinations can lead to large genomic rearrangements (LGR) encompassing the promoter region. The promoter region of the *BRCA1* gene is located between exon 1a and the first exon of the neighboring *NBR2* gene. Exon 1b (378 bp) is present in *BRCA1* transcripts essentially found in tumoral tissue, whereas exon 1a (181 bp) is found in normal tissue [6]. An Alu sequence AluSg is also located within exon 1b (Figure 1).

Most common pathogenic variants in *BRCA* genes have a small size, consisting of point mutations, small insertions or deletions in functional domains, or splice-site mutations, resulting in aberrant transcript processing. In 1997, LGR were first identified in *BRCA1* with two deletions: one within the coding sequence (1 kb deletion comprising exon 17 [7] and one involving the 5′ region (a 14 kb deletion of exons 1a to 2) [8]. In 2009, 45 different large genomic rearrangements for *BRCA1* have been characterized worldwide, including deletions and duplications of one or more exons [9]. Currently, LGR represent a substantial proportion of pathogenic variants in the *BRCA1* gene; the LGR frequency affecting the *BRCA1* gene varies in populations ranging from 3 to 15% [1,10,11,12]. A large proportion of these rearrangements (79%) correspond to deletions of one to several exons including the complete deletion of the gene [13]. Duplications encompassing one or several exons represent around 8% of *BRCA1* LGR. The remaining LGR are large insertion, deletion-insertion and triplication. Most of the breakpoints of *BRCA1* rearrangements are within intronic Alu sequences which cover more than 41.5% of the genomic sequence of *BRCA1* [4]. The main mechanism, 75% in the reported rearrangements [13], is postulated to be unequal homologous recombination between two homologous Alu sequences. The size of large rearrangements varies from a 244 bp deletion of exon 5 to a 160,880 bp deletion of exons 1 to 22 [12,14,15,16].

The majority of LGR are pathogenic. LGR are widely distributed within the *BRCA1* gene [13,17]. The French UNICANCER Genetic Group, in its national variant database [1,18,19], reports 536 families with 88 distinct large rearrangements of the *BRCA1* gene. Several hotspots of LGR have been reported within the coding exons: exons 3 to 8 duplication, exons 3 to 16 deletion, exons 8 to 13 deletion, exon 13 duplication, exon 15 deletion, exon 20 deletion, and exon 22 deletion. Some of them have been characterized [13,20] and have been considered as founder pathogenic variants, such as exon 13 duplication [21]. In this database, nineteen of these LGR (33%), present in 83 families, involve the 5′ region, confirming at a national level that this *BRCA1* region seems to be a hot spot of recombination. Nevertheless, the region with exons 1a/b to 2 seems to be very sensitive to those events and is the site of 15% of all rearrangements (12/81) reported in the literature [5,12,17,22,23,24]. This poses the question regarding whether all those events are due to founder pathogenic variants or the hot spot of recombination. The 5′ region of *BRCA1* is the site of two homologous large segmental duplications of 14 kb (chr17:43116961-43131375 and chr17:43153707-43178103) that created the pseudogene Ψ*BRCA1* 28 kb upstream from the *BRCA1* gene and causes a recombination hotspot, as evidenced by several 37 kb large rearrangements which delete exon 1a to exon 2 [5]. In fact, eight families with a 36,934 bp deletion involving Ψ*BRCA1* are described in the literature [13]. There were three groups of deletion with slightly different breakpoints, but all were of the same exact size and apparently without Alu sequences involved [13]. Twelve LGR involving at least exons 1a and 2 have been previously reported [13]. The segmental duplication and the presence of numerous Alu sequences explain the difficulty in identifying breakpoints due to the high homology in the breakpoint regions with only a few nucleotides of divergence [5].

This study aims to identify and characterize the LGR presented in the French high breast/ovarian cancer risk population. A selection of 10 LGR was carried out on 20 families focused specifically on the first exons 1 and 2 of the *BRCA1* gene. The selection of the LGR with focus on the exons related to the segmental duplication. We reported, for the first time, a new tandem triplication of exons 1 and 2 in six families and its classification as well as nine different deletions, of which eight were novel.

## 2. Materials and Methods

### 2.1. Family Selection and Screening Methods

The analysis was performed on germline DNA collected and extracted from blood samples. In total, 20 families were selected from all regions of France with large rearrangements in the 5′ region of *BRCA1* germline DNA analysis detected with the MLPA, qPCR, or QMPSF techniques between 1995 and 2008, as previously described [10,25]. For those patients, informed consent was obtained and sufficient germline DNA was used for further analyses of breakpoint characterization.

### 2.2. Dedicated Zoom-In CGH Array

*BRCA1* dedicated CGH arrays have already been proposed [26,27]. To characterize large rearrangements involving the 5′ region of the *BRCA1* gene, a zoom-in CGH array was used. A microarray with 10,195 oligonucleotides was specially designed with home-designed oligonucleotides and validated oligonucleotides from Agilent Technologies (Santa Clara, CA, USA). Of these, 7088 oligonucleotides were located throughout the genome while 3107 oligonucleotides were specifically designed for the *BRCA1* gene and its flanking regions. In genome assembly Hg38, the region covered by the oligonucleotides was from chr17:42815830–43364904.

In the 5′ region covered from chr17:43124115–43364904, there were 939 oligonucleotides on 267 kb with a resolution of 284 bp per oligonucleotide. In the coding region, there were 1136 oligonucleotides on 78 kb with a resolution of 72 bp per oligonucleotide. In the 3′ region, there were 1032 oligonucleotides on 132 kb with a resolution of 128 bp per oligonucleotide. The analytical approach has been described elsewhere [26].

### 2.3. qPCR and Breakpoint Sequencing

LGR were confirmed with either gene dosage or directly by sequencing the breakpoint. The gene dosage real-time quantitative PCR (qPCR) approach used four sets of primers: two sets inside the deletion and one set on each border. The qPCR was performed on a LightCycler 480^®^ (Roche Diagnostics, Penzberg, Germany) with High-Resolution Melting Master^®^ containing ResoLight^®^ dye (Roche Diagnostics), as described previously [28], and the analysis of quantitative result used the ΔΔCp method [29].

Characterization of large rearrangements was performed by classical PCR and Sanger sequencing. As a first screening, samples were amplified with the primer described by Puget et al. (Lit1—37 kb) and Swenssen et al. (Lit6—14 kb) [5,8]. The sequencing of the breakpoint used primers on each non-deleted border of the deletion. The classical PCR approach was used. PCR products were analyzed on agarose gel and sequenced in both directions by using the PCR primers with the BigDye Terminator Cycle Sequencing Reaction Kit and an ABI Prism 3130XL automated sequencer (Applied Biosystems Waltham MS, US). The main primers are reported in Table 1. In cases in which a shortage of material prevented PCR or sequencing, the signal of the CGH array was used to estimate the average size of the deletion.

### 2.4. Molecular Combing

The molecular combing analysis needs high-molecular-weight genomic DNA. For this purpose, the DNA extraction was carried out from lymphoblastoid cell lines after immortalization by Epstein–Barr virus, as previously described [30]. To spatially characterize the LGR, a high-resolution BRCA1 Genomic Morse Codes (GMCs) was applied (described in the following patent US20130130246A1). This technology considerably improves the structural and functional analysis of DNA across the genome and is capable of visualizing multiple genomic regions at high resolution (in the kb range) in a single analysis. A GMC is a series of “dots or dashes” (corresponding to DNA probes with specific sizes and colors) and “gaps” (corresponding to uncolored gaps with specific sizes located between the DNA probes), designed to physically map and define, with a specific “signature”, a particular genomic region [31,32]. Molecular combing and statistical analysis of the results were performed in the *BRCA1* locus as previously described [32].

### 2.5. Haplotype Study

Six short tandem repeat markers across the *BRCA1* gene were genotyped for patients with similar large rearrangements detected by CGH array. The markers used were D17S579, D17S855, D17S1322, D17S1323, D17S1326, and D17S1327, with primers available on the UCSC database and the forward primer labelled with FAM dye. The PCR products were separated using an ABI Prism 3130XL capillary automated sequencer (Applied Biosystems Waltham MS, US). Data output was analyzed to determine the haplotype with GeneMapper software (Applied Biosystems).

### 2.6. RNA Studies

RNAs were available from two patients with the triplication. A heterozygous SNP on the coding region was identified to estimate the allelic imbalance and possibly confirm a potential impact of the event on the *BRCA1* transcription. Sanger sequencing was performed to estimate the presence of the heterozygous SNP.

### 2.7. Tumoral Studies

The BRCAness has been assessed with the approach described by Popova et al. [33]. Two tumoral samples with the triplication involving exon 2 of the *BRCA1* were studied on a Cytoscan array (Affymetrix, Santa Clara, CA, USA). Frozen tissues were extracted using the Quickgene 610-L automated system from FujiFilm (Courbevoie, France) according to the manufacturer’s instructions and calibrated to 50 ng/μL by UV spectrophotometric assay (Nanodrop, Thermo Fisher Scientific, Villebon sur Yvette, France).

### 2.8. Screening of the 5′ Region and Exons 1a and 1b

A screen for point mutations and small rearrangements was performed in the 5′ region and exon 1a/1b since this region was the minimal region of deletion for all the rearrangement reported here. From Institut Curie—Saint Cloud, 384 families with breast cancer predisposition but without pathogenic variant in the *BRCA1/2* gene were selected and their DNA samples were screened. The scanned region measured 1359 bp from chr17:43,124,509–43,125,867 (100% of bases, 100% of span encompassing the non-coding exon 1b and its AluSg region, exon 1a and *NBR2* exon 1). The amplification was performed in a classical PCR cycling on a thermocycler VERITI (Applied Biosystems) with Ampli-Taq Gold mix (Applied Biosystems). Detection was performed on a LightCycler 480^®^ (Roche Diagnostics, Penzberg, Germany) adding LCGreen (Idaho Technology Inc., Salt Lake City, UT, USA).

### 2.9. Identification of Sensitive Regions

All the estimated or precise breakpoint coordinates identified in this report and in different works were gathered in groups to describe a common sensitive region with a size inferior to 3000 bp. To assess the impact of Alu in recombination in this region, the density of Alu sequences in sensitive regions was computed with Repeat Masker (www.repeatmasker.org accessed on 22 June 2021) by dividing the number of Alu nucleotides by the total number of nucleotides in the region.

### 2.10. Nomenclature

All rearrangements are described with the same orientation as the *BRCA1* gene—telomeric to centromeric. The 5′ breakpoint thus has a genomic position inferior to the 3′ breakpoint. The nomenclature used was based on HGVS recommendation and the Hg38 for the chromosome 17 (NC_000017.9). The NG5930 position reported in Sluiter et al. [13] was translated in this nomenclature with the software http://www.mutalyzer.nl/2.0/, accessed on 25 October 2020. In genome assembly NCBI36/Hg18, there is a gap of 100,000 bp in the 5′ region of the *BRCA1* gene from 38,601,721–38,701,721. All the sizes reported for deletions involving this region were corrected using the genome assembly Hg38. Due to the very large rearrangements, the nomenclature recommended by Sluiter et al. was not used [13], and a nomenclature based on Hg38 was used due to ease of interpretation.

### 2.11. Variant Classification

Genetic, clinical, familial, and tumoral data for patients carrying the triplication variant were collected in collaboration with the French UGG (UNICANCER Genetics Group) network and within the framework of the COsegregation of VARiants in the *BRCA1/2* and *PALB2* genes clinical trial (https://clinicaltrials.gov/ct2/show/NCT01689584, accessed on 15 May 2021, [18,34,35,36,37]). The cosegregation likelihood ratio (LR) of pathogenicity was assessed using the statistical model developed by Thompson et al. [38,39]. When cancer validation was not obtained, the phenotype is considered as an unknown. In addition to the cosegregation LRs derived here, for each family the multifactorial LR includes LR co-occurrence [40], family history [41,42] and breast pathology LRs [43], as previously published [44]. LRs are combined to calculate posterior probabilities using a Bayesian formula [40,44]. Combined LRs of families sharing the same variant were multiplied.

Combined LRs = LR_coseg_ × LR_co-occurrence_ × LR_Fam-Hist_ × LR_Pathology_.

Posterior Odds = Combined-LRs × [Prior Probability/(1—Prior Probability)].

Posterior Probability of pathogenicity = Posterior Odds/(Posterior Odds + 1) [44,45].

Given that the research question was to assess the clinical relevance of this molecular aberration (whether it is pathogenic or not), all variants had a prior probability of 0.5 [34,46,47].

## 3. Results

There have been 88 LGR of *BRCA1* in 536 families registered in the French database [1]. A selection of 10 of them, affecting the 5′ region of *BRCA1* and exons 1 and 2, has been characterized in 20 families (Table 2A,B, Appendix A). These 5′ *BRCA1* region large genomic rearrangements correspond to one deletion involving only non-coding exons 1a–1b, 8 deletions of the 5′ region including the first coding exon 2 and ATG, and one triplication of the 5′ *BRCA1* region. While one deletion (F5) was previously reported in the literature involving the region associated to the *BRCA1* pseudogene [5], 9 of these 10 LGR have never been reported and were precisely characterized. The positions of those events detected in the 20 families are shown in Figure 2. Neither familial history of probands carrying these LGR nor the tumor characteristics significantly differ from other high breast and ovarian cancer risk families (Appendix A).

### 3.1. Identification of a Triplication of BRCA1 5′ Region

An apparent duplication in the 5′ region detected either by MLPA or qPCR was identified in four different families but an abnormal profile presenting a signal intensity gain of more than two compared to the control DNA has been observed for the proband of family 15 (F15) (Appendix A). This high value of patient/control ratio suggested that this proband might harbor more than two copies of the *BRCA1* 5′ end. To further characterize this CNV, three probands carrying this apparent duplication have been submitted to molecular combing using high-resolution BRCA1 Genomic Morse Codes (GMCs). GMC revealed a triplication of the S7 probe green signal and part of the SYNT1 probe red signal, respectively, hybridizing with the 5′ region of *NBR2*, exons 1 and 2 of *BRCA1* for S7 and intron 2 for SYNT1. In contrast, the blue signal of the adjacent probe S6 was not modified (Figure 3). This result demonstrates that the copy number variation abnormality identified in these Hereditary Breast and Ovarian Cancer (HBOC) families is a triplication of the exons 1 and 2 and part of intron 2 of *BRCA1,* resulting in an insertion of about a 16.3 kb assessed with GMC probes. The characterization of the three rearrangements identified the same 7.3 kb tandem triplication of region localized on g.chr17:43,120,714–43,127,972 (Hg38).

The family histories of these families reveal classical characteristics of HBOC risk family without specific common particularity (Appendix A). All those families have been screened and selected with the criteria of HBOC.

To further investigate the consequence of this CNV (Copy Number Variation) on breast/ovarian cancer risk, we tended to determine the functional consequences of the presence of this CNV by analyzing tumoral mRNA stability and tumor profiles. An SNP was found to be heterozygous on tumoral mRNA, which is in favor of no impact on transcription and no loss of heterozygosity in the tumor (Appendix A). We then examined whether the available tumors of two probands from families F16 and F17 carrying this triplication entered the BRCAness type, as defined by Popova et al. [33]. As shown in Table 3, none of these tumors presented a BRCAness profile. These results strongly suggest that this up-to-date yet never reported triplication has low or no influence on *BRCA1* expression and tumorigenesis.

For all reported deletions, the impact was at least a loss of the promoter region, resulting in a putative absence of expression and leading to them being considered as pathogenic variants. For duplications, in the absence of additional evidence, the variants were classified as variants of unknown significance (VUS) in the first instance [54]. A cosegregation analysis was then performed on the families with the triplication variant. As shown in Table 4, this triplication variant had posterior probabilities of pathogenicity <0.001, classified as benign (class 1) and providing convincing evidence that this variant is not associated with a cancer risk.

### 3.2. Characterization of Different Deletions in the BRCA1 5′ Region

We then focused our attention on the eight unknown deletions of *BRCA1* 5′ region. These deletions removed different *BRCA1* segments whose sizes ranged from 1.4 to more than 37 kb (Figure 2 and Table 2).

A 1407 bp deletion with exact same breakpoint had been detected in F6 and F7, which suppressed the ATG start codon element in exon 2 and were classified as pathogenic by the UGG BRCA network [1]. With these two new rearrangements, four additional breakpoint positions previously reported (Lit1 to 4), would fall within a range of 1 kb region of intron 1b (sensitive region A2) [49,50].

Five families (F8, F9, F10, F11, F12) had a large deletion (estimated around 4–8 kb) and characterization of all five revealed three distinct large genomic rearrangements (Figure 4). Three families (F8, F9, F10) had similar haplotypes with the same 6757 bp deletion, suggesting a founding effect, while F11 had a smaller deletion of 6650 bp and F12 presented a deletion of 7248 bp. In contrast to the previously described rearrangement, Lit5, which includes an insertion [24], there was no insertion reported in these three rearrangements. The breakpoint of those events was very close to the duplication breakpoint.

A larger deletion of approximately 14 kb in length from exon 1 to exon 2 has been detected in a single family F13 (deletion size at 13,408 kb), in a region already reported as deleted (Lit6; [8]).

Five families (F1, F2, F3, F4, F5) had a common 37 kb deletion involving the *BRCA1* pseudogene, Ψ*BRCA1.* PCR performed with the primers described in the literature [5] were conclusive in two families, F1 and F5, which had a 36,934 bp deletion (Table 2). While the F5 family had a breakpoint similar to the one described by Puget et al. [5], F1 breakpoint was different from the one reported in the literature. F2, F3, and F4 breakpoints were similar with a 36,770 bp deletion and had similar haplotypes distinct from the ones of F1 and F5. This last deletion is the first one reported which differs from the 36,934 bp deletion and breakpoints previously reported (Lit1 and Lit2) and is presented here in F1 and F5 (Appendix A).

Finally, a deletion affecting exons 1a–1b and the promoter region yet without reaching the start codon was present in a single family F14 (exact deletion size 5568 bp). F14 deletion was at 0.8 kb from the ATG in exon 2 (sensitive region A2), and the breakpoint was in the AluSg sequence within exon 1b (sensitive region A4). Since RNA was not available for this patient, we could not determine whether the deletion had a deleterious effect. As shown in Table 4, all these deletions are classified as pathogenic.

All these data confirmed that *BRCA1* 5′ region presents loci that are highly sensitive to potential rearrangements.

### 3.3. Identification of Five Regions Highly Sensitive to LGR in BRCA1

A sensitive region that could be a hotspot of recombination is defined as a region with a large number of breakpoints. Until this study, the sensitive region of the *BRCA1* locus was a region including the *BRCA1* pseudogene and the *BRCA1* exon 2 locus [17]. Based on this study and other prior studies, all the estimated or precise breakpoint coordinates were gathered in groups in order to describe a common sensitive region with a size inferior to 3000 bp. Alu sequences, which represent a frequent recombination site, have been shown to represent 48.4% of the genomic sequence of the *BRCA1* gene (39,265 bp of Alu sequences in 81,155 bp *BRCA1* gene). The 5′ region to the ATG codons on exon 2 contains only 21% of Alu sequences (56,333 bp of Alu sequences in 268,140 bp of 5′ region—sensitive region A2). Our data, along with previously reported LGR, allowed us to gather the coordinates of 36 *BRCA1* breakpoints within the 5′ region to intron 2. We thus identified six sensitive regions (A0 to A5) of 0.4 to 2.2 kb in size with more than 50% of repeated sequences, except in region A2. Thirty-three of the 36 identified breakpoints (92%) within the 5′ region to intron 2 were located in these six sensitive regions (Table 5 and Figure 2).

As several breakpoints fall in exons 1a and 1b, and this region is the location of the smallest deletions of this study, we then searched variants in this region of 1359 pb in a population of 384 patients with breast cancer predisposition for which no mutation has been detected in the coding regions of *BRCA1* or *BRCA2*. No pathogenic variant or small deletion was detected in the HRM screening, indicating an absence of significant variants in this region.

Taken together, our results clearly demonstrate that the *BRCA1* 5′ region is highly sensitive to large rearrangements such as deletions and also triplications due to the presence of five sensitive regions between the *BRCA1* pseudogene and exon 2.

## 4. Discussion

The present study provides a comprehensive analysis of all large rearrangements involving the high LGR sensitive 5′ region of the *BRCA1* gene detected in eight French laboratories. A triplication and nine large deletions in the 5′ region were identified in 20 families.

Even if the largest proportion of *BRCA1* LGR is a deletion of one or several genes, duplications of some exons such as exons 3–8, 13, 18–20 or an interlocus gene conversion of *BRCA1* have been published in recent years due to the increasing use of easy screening techniques such as MLPA [55]. The high prevalence of some of these duplications is due to the founder effect, such as the exon 13 duplication which appears to originate from the UK [21]. More recently, a tandem repetition of *BRCA1* exons 1 and 2 through duplication of the 7259 nt sequence with the same breakpoints has been reported in a Spanish family [52]. The reports of triplication of the breast/ovarian cancer predisposition genes are very rare. The only triplication of *BRCA1* exons described up to now covering exons 17 to 19 was only found in a small family presenting a woman diagnosed for breast carcinoma at 31 years old [10]. A triplication of exons 13 to 24 of the *BRCA2* gene has also been reported [56]. In this study, we identified and characterized, for the first time, a tandem triplication of the *BRCA1* 5′ end concerning exons 1 and 2 and part of intron 2 in high-risk breast/ovarian cancer families. The presence of this triplication does not affect mRNA transcription, showing that there is no loss of heterozygosity on the normal allele and that triplication does not impact the expression. Furthermore, the genomic profile of the two corresponding tumors is not similar to the BRCAness profile defined by Popova et al., suggesting that this variant may have low or no impact on BRCA1 function [33]. We conducted a complementary analysis with the multifactorial model which allowed us to classify this triplication as benign and not related the cancer predisposition in those six families.

Deletions occurring in the *BRCA1* 5′ region involve various parts of the gene and of the neighbor genes, *NBR2* and *BRCA1* pseudogene *ΨBRCA1*. Of the 14 families carrying a large deletion, only family F5 had a deletion previously reported in the literature. In this French population, at least three recurrent deletions were identified: 1.4 (F6, F7), 6.7 (F8, F9, F10), and 36.8 kb (F2, F3, F4). The founding effect of those rearrangements was confirmed with haplotype analysis and could reflect founding mutations.

Our mapping of 36 breakpoints led us to identify six sensitive regions; all those regions define sensitive points which could be specifically explored. This study could help to better design an NGS strategy for the detection and characterization of those events [57,58]. Since the number of sensitive regions is limited, there should be both the implication of the Alu sequence and chromatin conformation. This could be illustrated with Figure 5, which is a proposed structural interaction to explain the non-allelic homologous recombination leading to the 36,934 bp deletion. This size is precisely the position between *BRCA1* and *ΨBRCA1*. All those sensitive regions could help to further understand the conformation of the chromatin loop, which can favor genomic instability in the locus of the *BRCA1* gene.

Future research should be driven by these results to improve the analytical process to identify those rearrangements for screening of both germline and somatic mutations. The very high susceptibility to rearrangement could reveal some complex events such as intragenic inversion. Globally, this could be an investigation pathway into some missing hereditability familial cases.

## 5. Conclusions

In conclusion, our study confirms the hypothesis of the high degree of susceptibility of the 5′ region of the *BRCA1* gene to large genomic rearrangements. Several large rearrangements described here had never been characterized in the literature. These results underline the importance of LGR analysis in the mutation screening of high-risk breast/ovarian cancer family with particular attention to the analysis of the highly LGR sensitive 5′ region of *BRCA1*.

## Figures and Tables

**Figure 1 cancers-13-03171-f001:**
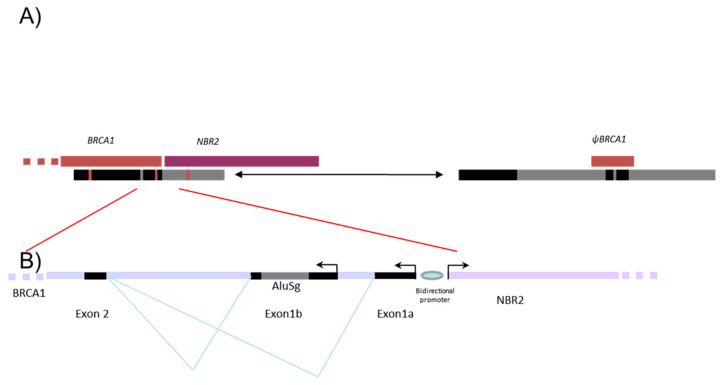
Schema representing the 5′ region of *BRCA1* gene. Description of the 5′ region of the *BRCA1* gene including *NBR2* gene (purple) and *BRCA1* pseudogene (Ψ*BRCA1*) (dark red). (**A**) The positions of the two segmental deletions shown in the gray and in black boxes are the homologous regions with more than 80% identity. (**B**) Alu sequences are indicated as gray line.

**Figure 2 cancers-13-03171-f002:**
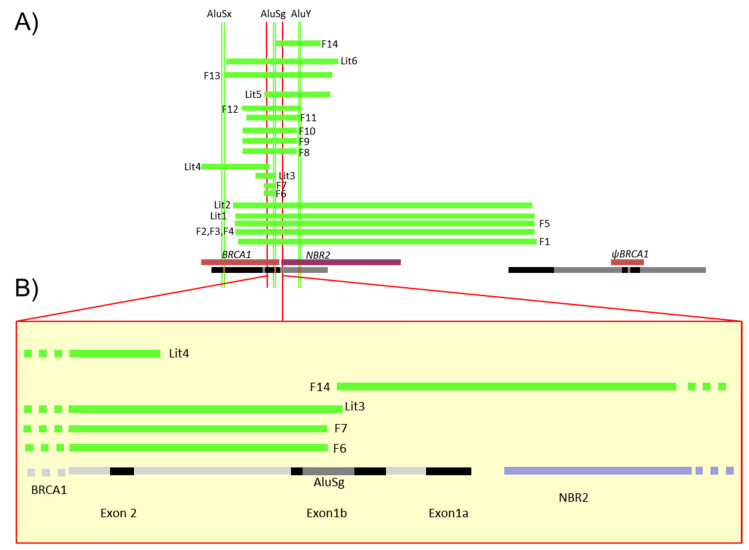
Schema of the different LGR detected in French families. (**A**) This diagram gives the description of Table 2 large deletions detected in the 5′ region described in 14 families. Green lines represent deletions detected in the French families (F1 to 14) or previously reported in the literature (Lit1 to 6); all segments are represented proportionally. At top, the Alu sequences (AluSx, AluSg, AluY) were also positioned precisely to identify specific breakpoints. At bottom, the *NBR2* gene is in purple, *BRCA1* and *pseudoBRCA1* in brown and segmental duplication in black. (**B**) Focus on the 5′ region of *BRCA1* including exon 2, non-coding exons 1a and 1b, the AluSg position, bidirectional promoter, and first exon of the *NBR2* gene in black, *BRCA1* introns in grey and *NBR2* introns in purple.

**Figure 3 cancers-13-03171-f003:**
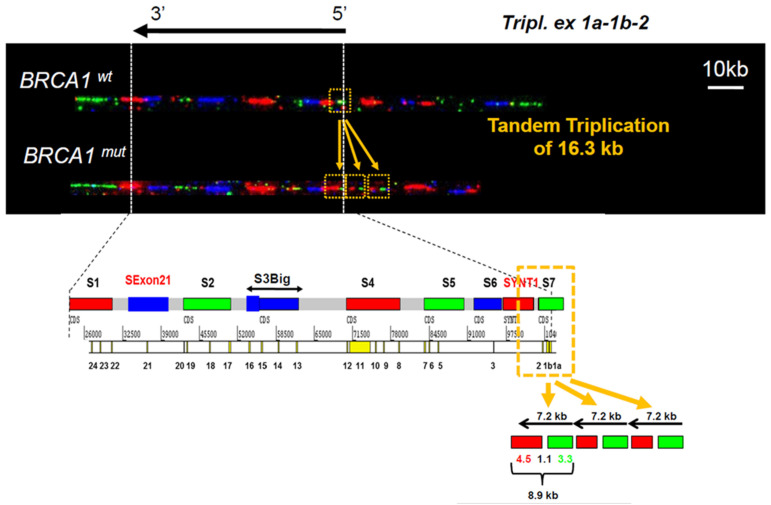
Molecular combing analysis of *BRCA1* 5′ region. Molecular combing was performed using high-resolution BRCA1 Genomic Morse Codes, as previously published [32]. The complete BRCA1 GMC covers a genomic region of 200 kb and is composed of 9 probes (S1–S7) of a distinct color (green, red, or blue). Triplication of exons 1 and 2 and part of the intron 2 of *BRCA1* gene is visible as a tandem repeat triplication of the red and green signal of S7 and SYNT1 probes. Data are representative of the analyses performed on DNA from families F15, F16, F17, and F18. The sizes are estimated with this technology and are not defined precisely contrary to Sanger sequencing.

**Figure 4 cancers-13-03171-f004:**
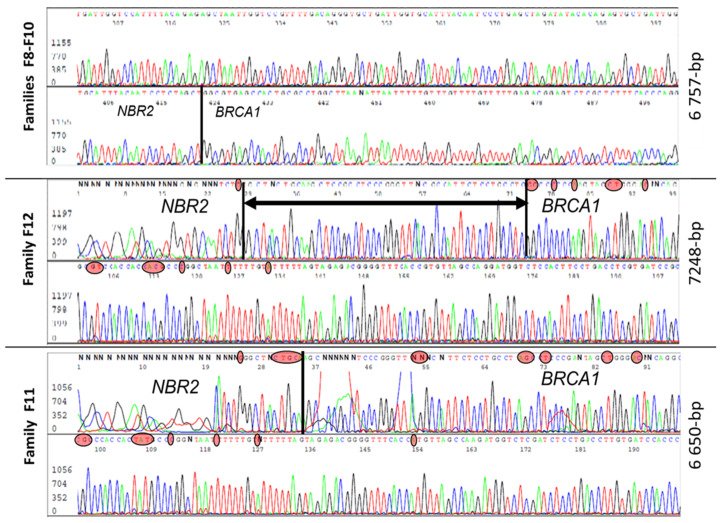
Sanger sequences analysis of three different 5–10 kb deletions with very close breakpoints harbored by individuals from five unrelated families. In red circles are nucleotides specific to the *BRCA1* and *NBR2* genes. The black vertical line is the breakpoint position and the horizontal line with arrows is a common region in which there is a breakpoint.

**Figure 5 cancers-13-03171-f005:**
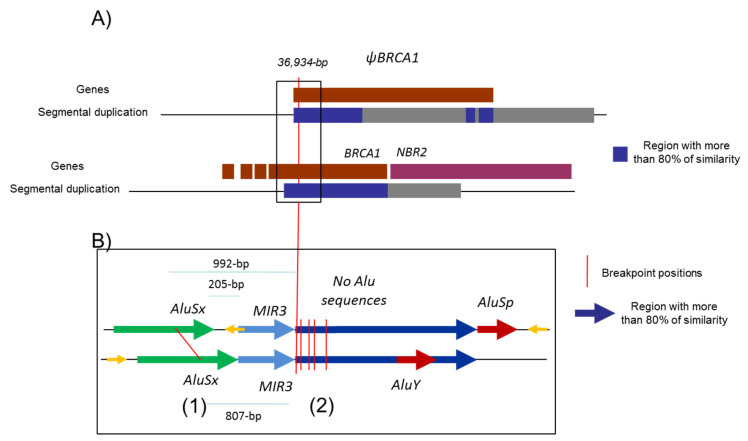
Structural organization of *BRCA1* 5′ region. (**A**) Proposed structural interaction to explain the non-allelic homologous recombination which leads to the 36,934 bp deletion reported here (in F1 and F5 families) and in the literature (Lit1, Lit2). *BRCA1* and *pseudoBRCA1* are represented in brown and *NBR2* in purple. Under the genes, blue and grey boxes represent segmental duplications, with the blue box presenting more than 80% similarity. There is likely a chromatin loop which facilitates the recombination by bringing the two loci close together. (**B**) Analysis of the similarity of ΨBRCA1 and BRCA1 gene in the region with the reported breakpoints—chr17:43155400–43157973 and chr17:43118270–43121226. (**1**) Breakpoints identified in F2, F3, F4 and their distance to the segmental duplication with high homology (blue array); (**2**) breakpoints of F1, F5, Lit1 and Lit2.

**Table 1 cancers-13-03171-t001:** Primers used in the characterization of the detected deletions and in the screening of the exon 1 locus.

Family	F (5′-3′):	R (5′-3′):
F8, F9, F10, F11, F12	TCCCTAACCAAGATCTCTTTAAC	CTAAAACAGGCTGAAAACCTTAC
F1, F5	ACCTAAAATTCCTTCTGCTGGAC	TTGGATAAAGTCCAGGCAAGAT
F2, F3, F4 *	CTTCAGAAAATACATCACCCA	ATAAGGTTACTGTCCCCGA
F14	TCAAAGGATGAGTTGGGCATA	AAGGTCAATTGTGTTCATTTGCAT
Lit6	CCACTGAGGACCTAAAGCATAA	GATATTGTAGGGAAAGACTATCAG
F13	ATGAGTATGGGGCTAAGACA	CATTTGACCTGTGGAGTTTC
F6, F7	AGACTTCCTGGACGGGGGAC	TAAGGAACACTGTGGCGAAGA
Exon 1 screening	CCTAACCTTTCCCAGTGACCTG	TAAGGAACACTGTGGCGAAGA
F15, F16, F17, F18	GTTCAAGTTCAAGCGCTTCTC	GTGTCTAGCTTGGGGTTTGG

* Primer for sequencing: GCAGAGGCGGGGTTTCCCCATATTGGCTAGGCTGGTCTCA.

**Table 2 cancers-13-03171-t002:** Description and characterization of large genomic rearrangements involving the 5′ region of the *BRCA1* gene. A/ Families with deletion; B/ Families with triplication. A0, A2, A3, A4, A5: sensitive region with several breakpoints reported, S: breakpoint in the segmental duplication involving *BRCA1* intron 2 and *BRCA1* pseudogene, HS: breakpoint outside the segmental duplication, NE: not explored in this study. S: specific breaking point region; Lit: Family of literature. * Class 5 = pathogenic [48].

Family	Deleted Exons (MLPA, QMPSF)	Frequency	Estimated Size in CGH Array (kb)	Size of the Deletion (bp)	Deleted Locus in (From 3′ to 5′ of the *BRCA1* Gene) chr17.hg38	Sensitive Regions in Which the 5′ and 3′ Breakpoints Are Localized	Class	Reference
A/								
F1	Exons 1a-2 ~38 kb	1	33–38	36,934	g.43120187_43157120del	A1–A5	5 *	This report
F2 F3 F4		3	33–38	36,770	g.43118776_43155546del	A0–A5	5	This report
F5		1	33–38	36,934	g.43119898_43156832del	A1–A5	5	This report and in [5,12,23,24]
Lit1		-	-	36,934	g.43119950_43156884del	A1–A5	5	[5]
Lit2		-	-	36,934	g.43119629_43156562del	A1–A5	5	[17]
F6 F7	Exon 2 <1 kb	2	<2	1407	g.43123453_43124859del	A2–A2	5	This report
Lit3	Exon 2	-	-	2535	g.43122409_43124943del	S–A2	5	[49]
Lit4	Exons 2–3	-	-	9113	g.43115107_43124219del	A2–S	5	[50]
F8 F9 F10	Exons 1a–2 ~5–10 kb	3	4–8	6757	g.43120817_43127573del	A1–A3	5	This report
F11		1	4–8	6650	g.43121312_43127961del	A1–A3	5	This report
F12		1	4–8	7248	g.43120725-43127972del	A1–A3	5	This report
Lit5	Exons 1–2 ~5–10 kb	-		8168	g.43123549_43131716delinsAAAAAAAAA	A2–A4	5	[24]
F13	Exons 1a–2 13–14 kb	1	13–14	13,408	g.43118518_43131925del	A0–A4	5	This report
Lit6		-	-	13,815	g.43118829_43132643del	A0–A4	5	[8,51]
F14	Exon 1a–1b	1	4–6	5568	g.43124936_43130504del	A2–A4	5	This report
B/								
**Family**	**Triplicated Exons** **(MLPA)**	**Frequency**	**Estimated Size in CGH Array (kb)**	**Size of the Fragment Triplicated (bp)**	**Triplicated locus in** **(From 3′ to 5′ of the *BRCA1* Gene)**	**Sensitive Regions in Which the 5′ and 3′ Breakpoints Are Localized**		**Reference**
F15, F16, F17, F18, F19, F20 Lit7	Exon 1–2	7	6–8 kb	7259	g.:43,120,714–43,127,972	A1–A3		This report and [52,53]
Lit8	5′UTR	1	-	44,430	g.43,124,439–43,168,868	A2–S		[54]
Lit9	Exon 1	3	-	36,915	g.43,119,759–43,156,673	A1–A5		[54]

**Table 3 cancers-13-03171-t003:** Tumor analysis and LST.

	Tumor 1 (F16)	Tumor 2 (F17)
**Family**	**F16**	**F17**
**Cellularity**	60%	50%
**LST score**	18	5
**Tumor ploidy**	4	2
**Tumoral cells**	46%	45%
**Conclusion**	LST low, no evidence of BRCAness	LST low, no evidence of BRCAness

**Table 4 cancers-13-03171-t004:** Classification of LGR affecting the 5′ region of *BRCA1* and exons 1 and 2.

Variant	Prior Probability	LR Cosegregation	LR Pathology	LR Family History	LR Combined	Posterior Probability of Pathogenicity	Classification
**Exons 1a–2** ~38 kb	0.5	119.105	15.4049	2.4	4403.52147	0.99977296058	5 (Pathogenic)
**Exons 1a–2** ~5–10 kb	0.5	118.28357	3.377408	4.170588	1666.11616	0.99940016	5 (Pathogenic)
**Triplication**	0.5	0.060189299	0.006830208	0.786824	0.000323467	0.00032336283	1 (Benign)

**Table 5 cancers-13-03171-t005:** Description and size of the five sensitive regions with hotspot of recombination in the rearrangements reported in this report and in the literature.

Locus	Position	% Alu Sequence	Size Pb	Segmental Duplication and Repeated Sequence	Number of Breakpoints
A0	43118500–43118900	94%	400	AluJr4-AluSz	3
A1	43120150–43121400	68%	1250	AluYk3-AluSg-AluYm1	9
A2	43123400–43125000	31%	1600	AluSc5-[Exon 2]-AluSx	7
A3	43127500–43128000	100%	500	LTR12C-AluY	4
A4	43130500–43132700	59%	2200	AluSz-MIRc-LIME3C2-[NBR2 exon]-AluY-AluSz6-MIRc-AlSc	4
A5	43155500–43157200	50%	1700	AluJr4-AluSz-AluSx-MIRc	6

## Data Availability

All LGR have been submitted to the French UMD-BRCA1 database (http://www.umd.be/BRCA1/, assessed on 15 October 2020).

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
