# Peer review of "5′ Region Large Genomic Rearrangements in the BRCA1 Gene in French Families: Identification of a Tandem Triplication and Nine Distinct Deletions with Five Recurrent Breakpoints"

_cancers, 2021, doi:10.3390/cancers13133171_

Round 1

Reviewer 1 Report

Caputo et al., report the identification and characterization of  LGRs identified in the French high breast / ovarian cancer risk population. The authors selected 20 families  with large rearrangements in the 5 'region of BRCA1 from French UMD-BRCA1 / BRCA2 database. A selection of 10 of them, nine deletions and one triplication, affecting the 5' region of BRCA1 and exons 1 and 2, has been characterized. This work confirms the hypothesis of the high degree of susceptibility of the 5 'region of the BRCA1 gene to large genomic rearrangements and also comes to the classification and characterization of a triplication as benign, by examining tumor tissue of 2 proband carriers of this CNV.
The authors confirm that BRCA1 5 'region presents loci highly sensitive to potential rearrangements and suggest the possibility of using the tumor tissue to be included in the multifactorial analysis for the classification of CNVs.
My opinion is that from a methodological point of view the paper is written in a clear and linear manner and must be considered for publication. This paper opens the discussion on different scenarios that must be considered when CNVs are identified.

Minor point: It would be useful to include in the paper or in supplementary a table with the classification of all the variants analyzed including the multifactorial analysis that led to the attribution of the clinical class, both for the new variants characterized and for those reported on UMD.

Author Response

Reviewer 1:

Caputo et al., report the identification and characterization of LGRs identified in the French high breast / ovarian cancer risk population. The authors selected 20 families with large rearrangements in the 5 'region of BRCA1 from French UMD-BRCA1 / BRCA2 database. A selection of 10 of them, nine deletions and one triplication, affecting the 5' region of BRCA1 and exons 1 and 2, has been characterized. This work confirms the hypothesis of the high degree of susceptibility of the 5 'region of the BRCA1 gene to large genomic rearrangements and also comes to the classification and characterization of a triplication as benign, by examining tumor tissue of 2 proband carriers of this CNV.
The authors confirm that BRCA1 5 'region presents loci highly sensitive to potential rearrangements and suggest the possibility of using the tumor tissue to be included in the multifactorial analysis for the classification of CNVs.
My opinion is that from a methodological point of view the paper is written in a clear and linear manner and must be considered for publication. This paper opens the discussion on different scenarios that must be considered when CNVs are identified.

Minor point: It would be useful to include in the paper or in supplementary a table with the classification of all the variants analyzed including the multifactorial analysis that led to the attribution of the clinical class, both for the new variants characterized and for those reported on UMD.

Answers to the Comments of Referee 1:

All the variants are reported in the French database. The multifactorial analysis was only performed on the triplication variant as it was reported as a class 3 variant. For all the other deletions, the classification in the French database was considered as class 5 as pathogenic variant and no other investigation was performed. The classification has been reported in table 1A and we added multifactorial analysis when it was possible in table 3.

Reviewer 2 Report

Caputo et al.,

Large genomic rearrangements (LGR) can be missed by standard sequencing techniques but can be the source of HBOC syndrome. This study identifies 10 new LGRs in the BRCA1 5’ region. 9 LGRs are expected to be pathogenic and 8 were never reported before. The authors show that the 5’ region is hotspot for LGR given the abundance of Alu sites.

Overall this is a nicely performed study that is important for the field and patients with HBOC syndrome. The work suggests that more effort should be given to detection of LGRs in patients with HBOC.

I believe that this study is ready for publication with some small edits and clarifications.

  1. It was not clear, or I missed it, was sequencing performed on tumor or normal DNA of these patients? Can this be made clearer in the results? If this was done in tumors, can the authors confirm these mutations in the germline? And vice versa?
  2. Was LOH assessed, can the authors mention or discuss this?
  3. Figure 2 and 5, I did fully understand, can more description be added to the legend to explain or the diagrams given additional detail?
  4. grammatical errors are present throughout and should be improved before publications.
  5. Line 414 – chromatine should this be chromatin
  6. Could the authors comment on whether other parts of the BRCA1 gene could also be hotspots for LGR? Only the 5’ region was studied here – however – what if the same amount of detail was given to the entire gene – would you find similar numbers of events in other parts?

Author Response

Reviewer 2:

Large genomic rearrangements (LGR) can be missed by standard sequencing techniques but can be the source of HBOC syndrome. This study identifies 10 new LGRs in the BRCA1 5’ region. 9 LGRs are expected to be pathogenic and 8 were never reported before. The authors show that the 5’ region is hotspot for LGR given the abundance of Alu sites.

Overall this is a nicely performed study that is important for the field and patients with HBOC syndrome. The work suggests that more effort should be given to detection of LGRs in patients with HBOC.

I believe that this study is ready for publication with some small edits and clarifications.

  1. It was not clear, or I missed it, was sequencing performed on tumor or normal DNA of these patients? Can this be made clearer in the results? If this was done in tumors, can the authors confirm these mutations in the germline? And vice versa?

Answers to the Comments of Referee 2:

Only two tumors with the triplication were sequenced to assess the LOH and LST score related to the pathogenicity. All other results are from germline DNA sequencing.

It is right that it can be more precise on the type of DNA used in this article. We then modify the paragraph:

“2.1. Families selection and screening methods

All the analysis was performed on germline DNA collected and extracted on blood samples. 20 families were selected from all regions of France with large rearrangements in the 5’ region of BRCA1 on germline DNA analysis detected with the MLPA, qPCR, or QMPSF techniques between 1995 and 2008 as described previously [10,22]. For all those patients, there were an informed consent and enough germline DNA to further analyses to breakpoint characterization. “

  1. Was LOH assessed, can the authors mention or discuss this?

Answers to the Comments of Referee 2:

The LOH was indirectly assessed on the RNA with a SNP. This approach validates the balanced expression of the two alleles showing that there is no loss of the normal allele and it does not impact the expression.

Page 12 line 399

“The presence of this triplication does not affect mRNA transcription and the genomic profile of the two corresponding tumors is not similar to the BRCAness one defined by Popova et al. suggesting that this variant may have low or no on BRCA1 function [30].”

Is modified as

“The presence of this triplication does not affect mRNA transcription showing that there is no loss of heterozygosity on the normal allele and the triplication does not impact the expression. Furthermore, the genomic profile of the two corresponding tumors is not similar to the BRCAness one defined by Popova et al. suggesting that this variant may have low or no impact on BRCA1 function [30].”

  1. Figure 2 and 5, I did fully understand, can more description be added to the legend to explain or the diagrams given additional detail?

Answers to the Comments of Referee 2:

We take into account your remark and describe better the figure in the legend.

Figure 2: Schema of the different LGR detected in French families. A) This diagram gives the description of Table 1 large deletions detected in the 5’ region described in 14 families. Green lines represent deletions detected in the French families (F1 to 14) or previously reported in the literature (Lit1 to 6); all segments are represented proportionnaly. In top, the Alu sequences (AluSx, AluSg, AluY) were also positionned precisely to identify specific breakpoints. In bottom, the the NBR2 gene is in purple , BRCA1 and pseudoBRCA1 in brown and segmental duplication in black. B) Focus on the 5’ region of BRCA1 including exon 2, the non-coding exons 1a and 1b, the AluSg position, the bidirectional promoter, and the first exon of the NBR2 gene in black , BRCA1 introns in grey and NBR2 introns in purple.

Figure 5. Structural organization of BRCA1 5’region. A/ Proposed structural interaction to explain the non-allelic homologous recombination which leads to the 36,934-bp deletion reported here (in F1 and F5 families) and in the literature (Lit1, Lit2). The BRCA1 and pseudoBRCA1 are represented in brown and NBR2 in purple. Under the genes, blue and grey boxes represent segmental duplications with blue box that present more than 80% of similarity. There is probably a chromatin loop which facilitates the recombination by getting the two loci close togehter. B/ Analysis of the similarity of ΨBRCA1 and BRCA1 gene in the region with the reported breakpoints – chr17:43155400-43157973 and chr17:43118270-43121226. (1) Breakpoints identified in F2, F3, F4 and their distance to the segmental duplication with high homology (blue array) (2) breakpoints of F1, F5, Lit1 and Lit2.

  1. Grammatical errors are present throughout and should be improved before publications.

Answers to the Comments of Referee 2:

The article has reviewed by an english speaker again to correct grammatical errors.

  1. Line 414 – chromatine should this be chromatin

Answers to the Comments of Referee 2:

We modified chromatine to chromatin

  1. Could the authors comment on whether other parts of the BRCA1 gene could also be hotspots for LGR? Only the 5’ region was studied here – however – what if the same amount of detail was given to the entire gene – would you find similar numbers of events in other parts?

Answers to the Comments of Referee 2:

The 5’ region is a very sensitive region in BRCA1 with a high concentration of segmental duplication and Alu sequences. There is no other region in the gene with such a recurrence. Sluiter, M.D et al 2011 have reviewed a large number of deletion and no other recurrence loci was identified. Most of the recurrent LGR reported in database and literature are documented as possibly founder events (i.e. exon 13 duplication). That is why, for the 5’region, the question is if all those events are founder mutation or due to hot spot of recombination.

Furthermore, we reported here in the same region both deletion and duplication. To clarify, we developed the paragraph page 3 line 106

“The majority of LGR are pathogenic. LGR are widely distributed within the BRCA1 gene [13,17]. The French UNICANCER Genetic Group reports in its national variant database [1,18,19], 536 families with 88 distinct large rearrangements of BRCA1 gene. Several hotspots of LGR have been reported within the coding exons: exons 3 to 8 duplication, exons 3 to 16 deletion, exons 8 to 13 deletion, exons 13 duplication, exon 15 deletion, exon 20 deletion, exon 22 deletion. Some of them have been characterized [13,20] and have been considered as founder mutations, as exon 13 duplication[21]. In this database, nineteen of these LGR (33%), present in 83 families, involve 5’ region, confirming at a national level that this BRCA1 region seems to be a hot spot of recombination. Nevertheless, the region with exons 1a/b to exon 2 seems to be very sensitive to those events and is the site of 15% of all rearrangements (12/81) reported in the literature [5,12,17,22–24]. This sets the question to know if all those events are founder mutation or due to hot spot of recombination.”

Reviewer 3 Report

Summary

In this study Caputo et al. characterize large genomic rearrangements (LGR) in BRCA1 in French high-risk-breast/ovarian cancer in families. The authors use dedicated CGH-array, high-resolution BRCA1-Genomic-Morse-Codes-analysis, and Sanger-sequence to identify the specific BRCA1-alterations, focusing on 20 families that have already been shown previously to have LGRs in BRCA1 5’ region. One duplication and 9 deletions are reported and the likelihood that these aberrations predispose an individual to breast and/or ovarian cancer is assessed.

It is undoubtedly critical to carefully characterize germline genomic aberration in BRCA1, distinguish between aberrations that lead to disease predisposition and those that have minimal to no impact, and use this information for disease management and monitoring. While this study identifies the specific BRCA1-LGRs that occur in a subset of at-risk families, it barely provides any characterization of the downstream functional implications of these LGRs (other than testing for BRCAness in 2 tumor samples). As it was already known that these families have large rearrangements in the 5’ BRCA1 region, the authors should provide a more in-depth characterization of these LGRs beyond mere identification. Below are specific comments along these lines with requests for clarifications and further investigation.

Major comments

  1. As the focus of the study is to characterize BRCA1 LGRs in the 5’ region, it is highly recommended that the authors invest in characterizing the aberrations they identified. What is the impact of these aberrations on the protein function, chromatin compaction, BRCAness, cell states, and resulting tumor tissue?
  2. BRCAness was tested only in 2 tumor samples, which is hardly enough to make a statically sound statement. The lack of BRCAness in these two samples does not necessarily mean that the triplication is nonfunctional. The authors should either test this hypothesis in additional tumor samples or considerably soften their conclusions.
  3. What is the added value of this study given that these families were already known to have LGRs in this region?
  4. The authors mention the impact on chromatin structure, but do not provide any data to support that.
  5. Under “2.4. Molecular Combing” the authors state that “Lymphoblastoid cell lines were immortalized by Epstein Barr virus as previously described.” How were lymphoblastic cell lines were used in this context?
  6. The multifactorial model that was used to determine the likelihood that a variant leads to breast and/or ovarian cancer predisposition should be described more comprehensively. Currently the method section mostly consists of references to previous papers where this model was used.

Minor comment: The color code in the legend of figure 1 does not match what is shown in the figure itself.

Author Response

Reviewer 3:

In this study Caputo et al. characterize large genomic rearrangements (LGR) in BRCA1 in French high-risk-breast/ovarian cancer in families. The authors use dedicated CGH-array, high-resolution BRCA1-Genomic-Morse-Codes-analysis, and Sanger-sequence to identify the specific BRCA1-alterations, focusing on 20 families that have already been shown previously to have LGRs in BRCA1 5’ region. One duplication and 9 deletions are reported and the likelihood that these aberrations predispose an individual to breast and/or ovarian cancer is assessed.

It is undoubtedly critical to carefully characterize germline genomic aberration in BRCA1, distinguish between aberrations that lead to disease predisposition and those that have minimal to no impact, and use this information for disease management and monitoring. While this study identifies the specific BRCA1-LGRs that occur in a subset of at-risk families, it barely provides any characterization of the downstream functional implications of these LGRs (other than testing for BRCAness in 2 tumor samples). As it was already known that these families have large rearrangements in the 5’ BRCA1 region, the authors should provide a more in-depth characterization of these LGRs beyond mere identification. Below are specific comments along these lines with requests for clarifications and further investigation.

Major comments

  1. As the focus of the study is to characterize BRCA1 LGRs in the 5’ region, it is highly recommended that the authors invest in characterizing the aberrations they identified. What is the impact of these aberrations on the protein function, chromatin compaction, BRCAness, cell states, and resulting tumor tissue?

Answers to the Comments of Referee 3:

All the variants are reported in the French database. This article have systematically characterized the breakpoint of all those alterations. The impact of these alterations was studied with tumoral tissue when available and multifactorial analysis.

For all the studied deletions, the classification in the French database was considered as class 5 as pathogenic variant. The deletion covered the traduction initiation site (ATG), the promoter and the non coding exons 1. This were enough to conclude on the deleterious impact of those variants. We do not produce any additional information on the protein as without any traduction site, we inferred that this is no protein. The multifactorial analysis was also performed for two recurrent deletions with the same size and the conclusion was highly significant to pathogeny. Tumoral tissues were not systematically available and no relevant analysis was possible.

For the triplication, the alteration was localized and in continuity. The tumoral mRNA analysis showed clearly the impact of the triplication as no variation in transcription was observed. The tumoral tissue. The multifactorial analysis was performed on the triplication variant as it was reported as a class 3 variant. The conclusion was clearly in favor to a neutral variant.

The classification was reported in table 1A and we added multifactorial analysis when it was possible in table 3. In the supplementary table 1, cosegregation data and pathology data were present. Chromatin compaction and cell state could be a specific additional work, but we do not have access to the methodology.

  1. BRCAness was tested only in 2 tumor samples, which is hardly enough to make a statically sound statement. The lack of BRCAness in these two samples does not necessarily mean that the triplication is nonfunctional. The authors should either test this hypothesis in additional tumor samples or considerably soften their conclusions.

Answers to the Comments of Referee 3:

The tumoral information on only 2 tumor samples is clearly not enough alone. We identified both a lack of BRCAness in these two samples but also a normal transcription of the two allele in breast cancer. We also have information of transcription by analyzing a tumoral mRNA. The paragraph page 9 was modified to clarify the results as followed:

“To further investigate the consequence of this CNV on breast/ovarian cancer risk, we tended to determine the functional consequences of the presence of this CNV by analysing tumoral mRNA stability and tumor profiles. A SNP was found heterozygous on tumoral mRNA which is in favour of no impact on transcription and no loss of heterozygosity in the tumor (supplementary figure 2).”

Finally, the multifactorial model was used to definitively classify the variant with cosegregation data (Table 3).

  1. What is the added value of this study given that these families were already known to have LGRs in this region?

Answers to the Comments of Referee 3:

The added value is first to better characterize the breaking point which were not precisely identify and to highlight the differences between all these LGRs which concern the same region.

  1. The authors mention the impact on chromatin structure, but do not provide any data to support that.

Answers to the Comments of Referee 2:

            The size of the LGR in families F1, F5, Lit1 and Lit2 is exactly a size of 36,934 pb (table 1) but the position were different from families F2, F3 and F4. The event was therefore totally different as well as the size but in the same region. For this information, we infer a chromatin structure with proximity of the segmental duplication to explain the strict conservation of the distance between different deletions. This information is only an inference related to the breakpoint positions and will need further investigations.

This is in the discussion page 13: “This could be illustrated with figure 5 which is a proposed structural interaction to explain the non-allelic homologous recombination leading to the 36,934-bp deletion. This size is precisely the position between the BRCA1 and ΨBRCA1. All those sensitive regions could help to further understand the conformation of the chromatin loop which can favor genomic instability in the locus of the BRCA1 gene.”

  1. Under “2.4. Molecular Combing” the authors state that “Lymphoblastoid cell lines were immortalized by Epstein Barr virus as previously described.” How were lymphoblastic cell lines were used in this context?

Answers to the Comments of Referee 3:

We need viable cells for the molecular combing to have high molecular weight genomic DNA after extraction. We modify the paragraph 2.4 page 5 as followed: “The molecular combing analysis needs high-molecular-weight genomic DNA. For this purpose, the DNA extraction was done from lymphoblastoid cell lines after immortalization by Epstein Barr virus as previously described.”

  1. The multifactorial model that was used to determine the likelihood that a variant leads to breast and/or ovarian cancer predisposition should be described more comprehensively. Currently the method section mostly consists of references to previous papers where this model was used.

Answers to the Comments of Referee 3:

We modified the paragraph 2.11:

“Genetic, clinical, familial, and tumoral data for patients carrying triplication variant were collected in collaboration with the French GGC network and within the framework of the COsegregation of VARiants in the BRCA1/2 and PALB2 genes clinical trial (https://clinicaltrials.gov/ct2/show/NCT01689584 [21,31–34]). Cosegregation likelihood ratio (LR) of pathogenicity is assessed using the statistical model developed by Thompson et al. [35,36]. When cancer validation is not obtained, the phenotype is considered as unknown. In addition to the cosegregation LRs derived here, the multifactorial LR includes for each family LR co-occurrence [37], family history [38,39] and breast pathology LRs [40] as previously published [41]. LRs are combined to calculate posterior probabilities using Bayesian formula [37,41]. Combined LRs of families sharing the same variant are multiplied.

Combined-LRs = LRcoseg x LRco-occurrence x LRFam-Hist x LRPathology

Posterior Odds = Combined-LRs x [Prior Probability/(1 − Prior Probability)].

Posterior Probability of pathogenicity = Posterior Odds/(Posterior Odds+1)[41,42].

Given that the research question was to assess the clinical relevance of this molecular aberration (whether it is pathogenic or not), all variants were a prior probability of 0.5 [43,44].

Minor comment: The color code in the legend of figure 1 does not match what is shown in the figure itself.

Answers to the Comments of Referee 3:

The color code was modified and legend was improved.